# Microstructure and Superplastic Behavior of Ni-Modified Ti-Al-Mo-V Alloys

Anton D. Kotov [1,*], Maria N. Postnikova [1], Ahmed O. Mosleh [1,2], Vladimir V. Cheverikin [1] and Anastasia V. Mikhaylovskaya [1]

[1] Department of Physical Metallurgy of Non-Ferrous Metals, National University of Sciences and Technology "MISIS", 4 Leninskiy Ave. 4, 119049 Moscow, Russia; sitkina.m@misis.ru (M.N.P.); cheverikin80@rambler.ru (A.O.M.); mihaylovskaya@misis.ru (V.V.C.); ahmed.omar@feng.bu.edu.eg (A.V.M.)

[2] Department of Mechanical Engineering, Shoubra Faculty of Engineering, Benha University, Cairo 11629, Egypt

[*] Correspondence: kotov@misis.ru; Tel.: +7-495-955-01-34

**Abstract:** The paper studies the influence of 0.5–1.8 wt.% Ni alloying on the superplasticity, microstructural evolution, and dynamic grain growth effect in a temperature range of 625–775 °C and room temperature mechanical properties of two-phase Ti-Al-Mo-V alloys. Due to a decrease in $\beta$ transus and an enhancement in the alloy diffusivity, an increase in Ni content significantly improved superplasticity. The Ni-modified alloys exhibited 1.5–3-fold lower flow stress, a 2.5–3-fold greater elongation to failure, and 1.4–1.7-fold higher strain rate sensitivity $m$ coefficient compared to the Ni-free alloy. An intermetallic $Ti_2Ni$ compound precipitated in the 1.8 wt.% Ni-modified alloy during low-temperature deformation at 700 °C and decreased superplastic properties. The Ti-4Al-3Mo-1V-0.1B alloy with 0.9 wt.% Ni exhibited a good combination of the superplastic behavior and room-temperature mechanical properties: an elongation to failure of 500–900% at a low-temperature range of 625–775 °C and constant strain rate of $1 \times 10^{-3}$ s$^{-1}$ and a yield strength of 885 MPa and ultimate tensile strength of 1020 MPa after pre-straining for 100% in a superplastic regime and strengthening heat treatment.

**Keywords:** titanium alloys; superplasticity; microstructural evolution; dynamic grain growth kinetics; strain-induced precipitation; mechanical properties



## 1. Introduction

Designing new lightweight, high-strength, and high-performance alloys for easy manufacturing processes is necessary for the transportation industry. Extensive efforts and huge budgets are dedicated to developing the alloys with a high strength-to-weight ratio and facilitating their manufacturing process. Among these alloys, titanium-based alloys exhibited an excellent combination of physical and mechanical characteristics, i.e., a high strength-to-weight ratio. However, the manufacturing process of complex-shape parts from titanium alloys is the main challenge for expanding its use due to a relatively low elastic modulus resulting from the spring-back effect [1–3]. The superplastic forming (SPF) method is a feasible approach for resolving and overcoming this challenge. The main disadvantage of SPF for Ti-based alloys is the requirement of the elevated forming temperature [4]. For many Ti-based alloys, a good superplastic formability can be achieved in a temperature range of 800 °C to 1030 °C [4–9]. The increased temperature is required to stimulate the diffusion-controlled deformation mechanisms, which provide superplasticity. At low temperatures, the dislocation creep and climb mechanisms dominate during the deformation and unstable flow, showing a low strain rate sensitivity coefficient of $m < 0.3$, occurs. An increase in the deformation temperature raises alloy diffusivity and the fraction of high-diffusive and ductile $\beta$ phase that improve superplasticity. Conversely,

higher temperatures increase the susceptibility to dynamic grain growth. Generally, for titanium alloys, the optimal superplastic characteristics may be attained at a 40–50% volume fraction of the $\beta$ phase [4,9,10]. Low forming temperatures, however, decrease the power consumption and forming cycle time, reduce the forming die wear and cost of die materials, increase the die life, and prevent the production of an alpha-oxide film on the surface of fabricated components [2,11]. Therefore, novel titanium alloys with low-temperature superplasticity are required.

Grain refinement helps to decrease a superplastic forming temperature [4,12–15]. Finer grains help to decrease the flow stress and to increase the strain rate sensitivity of flow stress owing to the facilitation of grain boundary sliding [4,13,15,16]. The grain size has a significant impact on flow stress control in the superplasticity region with a high $m$ [7,17,18]. The modification of the alloys' composition can enhance superplasticity [19–24]. In this case, the improvement in the superplasticity is attributed to: (1) decreasing the temperature of $\beta$ transus, (2) increasing the alloys' diffusivity [20,25–30], and refining the grains' onset of the forming process [31–33]. In this regard, alloying with $\beta$-stabilizing elements with high tracer diffusivity, such as Ni, Co, and Fe [34], is a promising way to decrease the forming temperature. Nickel exhibits the highest diffusivity in $\beta$-titanium, and alloying with Ni should facilitate the diffusion-controlled superplastic deformation mechanisms, grain boundary sliding (GBS), and its accommodation with dislocation and diffusional creep at lower temperatures [30].

A Ti-4Al-1V-3Mo alloy has a good combination of mechanical properties at room temperature and excellent superplastic behavior, but only at a high temperature of about 870 °C [1,35,36]. Low temperatures reduce the consumable power and SPF cost for the alloy. This research focuses on the improvement of the superplasticity of the Ti-Al-V-Mo alloy via alloying with high-diffusive Ni, which should enhance grain boundary sliding and provide low-temperature superplasticity. The alloys studied contain a minor B addition to improve both processing and mechanical properties due to grain refinement for as-cast and thermomechanical processed alloys [37–41].

## 2. Materials and Methods

The Ti-Al-V-Mo alloys with Ni addition in a range of (0–1.8 wt.%) and a small amount of boron were studied (Table 1). For the alloys' processing, pure metals Ti (99.99 wt.%), Al (99.99 wt.%), V (99.95 wt.%), Ni (99.95 wt.%), and Ti-50 wt.% Mo master alloys were used. Ingots 100 g in weight with dimensions of $50 \times 40 \times 10$ mm$^3$ were casted via argon arc melting and solidification in a vacuum furnace, Arc Melter ARC200 (ARCAST Company, Oxford, MS, USA), with a copper water-cooling mold.

**Table 1.** Chemical composition of the studied alloys (wt.%).

| Alloy | Al | Mo | V | Ni | B | Ti |
|-------|-----|-----|-----|-----|-----|------|
| 0Ni | 3.7 | 2.9 | 1.2 | - | 0.1 | Bal. |
| 0.5Ni | 3.6 | 2.8 | 1.1 | 0.5 | 0.1 | Bal. |
| 0.9Ni | 3.7 | 2.7 | 1.0 | 0.9 | 0.1 | Bal. |
| 1.8Ni | 3.8 | 3.1 | 1.1 | 1.8 | 0.1 | Bal. |

The ingots were thermomechanically processed as follows:

1. Homogenization annealing at 800 °C for 1 h;
2. Heating and annealing at 950 °C for 30 min (in the $\beta$-phase field [1,24]) in a vacuum furnace followed by water quenching;
3. Hot rolling was processed in the ($\alpha + \beta$)-phase field at a temperature of 750 ± 10 °C with a total reduction of 90%. The strain per pass was varied in a range of 10–33% with a decrease of 0.5–1.0 mm per pass. To ensure constant rolling temperature, the samples were soaked for 7–10 min in the furnace between passes. The modified alloy

with 1.8 wt.% Ni was additionally processed at a temperature of $700 \pm 10$ °C to ensure a similar $\alpha/\beta$ phase ratio to that of the other studied alloys. After the rolling process, the ingots were treated in a Kroll reagent (92% $H_2O$ + 3% HF + 5% $HNO_3$) for 30 min to dissolve the $\alpha$-phase layer.

The chemical and phase composition of the alloys and the microstructure evolution after each stage of thermomechanical processing were analyzed using a scanning electron microscope (SEM), Tescan Vega 3 LMH (Tescan Brno s.r.o., Kohoutovice, Czech Republic), equipped with an energy-dispersive X-ray spectrometer, X-Max 80 (EDS) (Oxford Instruments plc, Abingdon, UK). An electron backscattered diffraction (EBSD) (HKL NordlysMax electron backscatter diffraction detector) (Oxford Instruments plc, Abingdon, UK) technique was used to study the grain and subgrain structure evolution of the alloys. The scan area was $50 \times 50$ μm$^2$, and a step size of 0.15 μm was used. The samples were mechanically ground using abrasive SiC papers with a dispersion from 300 to 4000 and subsequently fine-polished with OP-U silica-based suspension (Struers APS, Ballerup, Denmark). To determine the phase composition, an X-ray diffractometer (XRD) analysis in a Bruker D8 Advance diffractometer (Bruker, Billerica, MA, USA) with Cu-K$_\alpha$ radiation was applied.

The grain size and volume fraction of the $\alpha$ and $\beta$ phases were measured using the SEM images for the vacuum-annealed samples at a temperature range of 625–850 °C for 30 min and subsequently quenched with water. The measured volume fraction was compared with those predicted by a Thermo-Calc (Thermo-Calc Software, Stockholm, Sweden) software (TTTI3) database. In addition, the $\beta$-transus temperatures of the investigated alloys were calculated by Thermo-Calc software and experimentally confirmed via the differential thermal analysis (DTA) using a Labsys Setaram Differential Scanning Calorimeter (SETARAM Instrumentation, Caluire, France).

The uniaxial tensile tests were carried out using a Walter Bay LFM-100 testing machine (Walter + Bai AG, Löhningen, Switzerland) with an argon atmosphere for superplastic deformation behavior examination. The crosshead motion was controlled via DionPro software (Walter + Bai AG, Version. 4.80, Löhningen, Switzerland). Dog-bone-type tensile test proportional specimens with a gauge section size of $14 \times 6 \times 1$ mm$^3$ were cut from the sheets along the rolling direction. Before starting the tensile tests, the samples were heated to superplastic deformation temperatures (625, 700, and 775 °C) and annealed for 30 min. The uniaxial tensile tests were performed using three methods: (1) test with a step-by-step decrease in strain rate, (2) test with a constant strain rate of $1 \times 10^{-3}$ s$^{-1}$, and (3) test with a 20% increase in strain rate above the nominal $1 \times 10^{-3}$ s$^{-1}$ during straining to study the strain-induced evolution of the strain rate sensitivity $m$ coefficient (according to ASTM E2448-11 standard).

The mechanical properties at room temperature were determined using a Zwick Z250 test machine (Zwick Roell Group, Ulm, Germany) for three samples per alloy. Alloys were pre-strained to 100% at a temperature of 775 °C with a constant strain rate of $1 \times 10^{-3}$ s$^{-1}$, and heat treatment included water cooling at 775 °C and subsequent aging at 480 °C for 16 h. The tested tensile samples had a gauge section with a size of $25 \times 6 \times 0.8$ mm$^3$.

## 3. Results and Discussion

### 3.1. Phase Composition Analysis

Figure 1 illustrates the Thermo-Calc polythermal section of the Ti-4Al-3Mo-1V-0.1B-xNi phase diagram (Figure 1a,b), DTA data (Figure 1c), and XRD spectrums (Figure 1d) for the studied alloys. The diffraction peaks of XRD analysis belonged to the $\alpha$ and $\beta$ phases, and the TiB phase, which agreed with the Thermo-Calc data. The $\beta$-transus temperatures determined by DTA analysis (Figure 1c) were similar to the theoretical temperatures calculated by Thermo-Calc (Figure 1a). The $\beta$-transus temperature decreased with an increase in Ni content. According to the polythermal cross-section, the $\beta$ transus for the 0Ni, 0.5Ni, 0.9Ni, and 1.8Ni alloys were 920 °C, 910 °C, 900 °C, and 887 °C, respectively (Figure 1a). According to the DTA, the $\beta$-transus values were 922 °C, 909 °C, 897 °C, and 890 °C, respectively (Figure 1c).

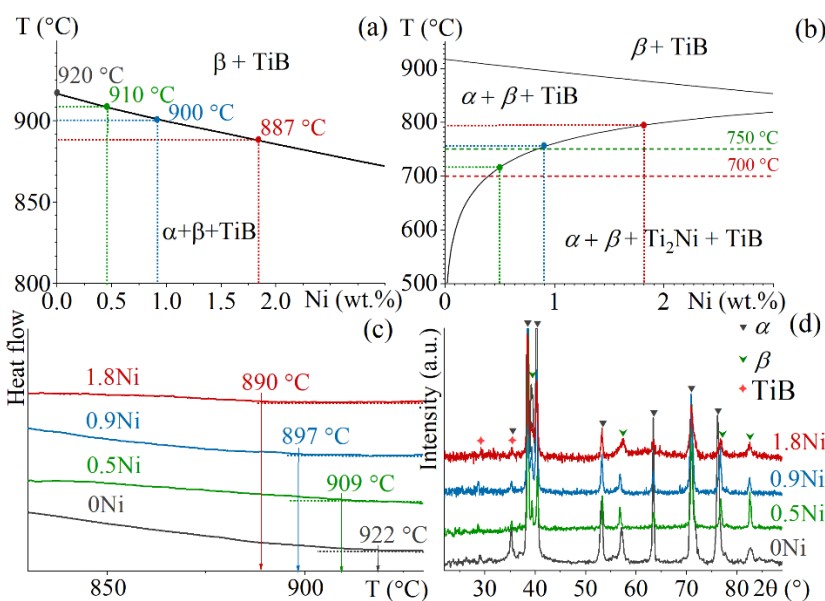

**Figure 1.** (**a**,**b**) Polythermal sections of the Ti-4Al-3Mo-1V-0.1B-xNi phase diagram constructed by Thermo-Calc software, (**c**) DTA spectra, and (**d**) XRD patterns for the investigated alloys.

### 3.2. Analysis of the Microstructure after Sheet Processing and Annealing

The α and β phases were elongated to the rolling direction after the thermomechanical treatment (Figure 2a–d).

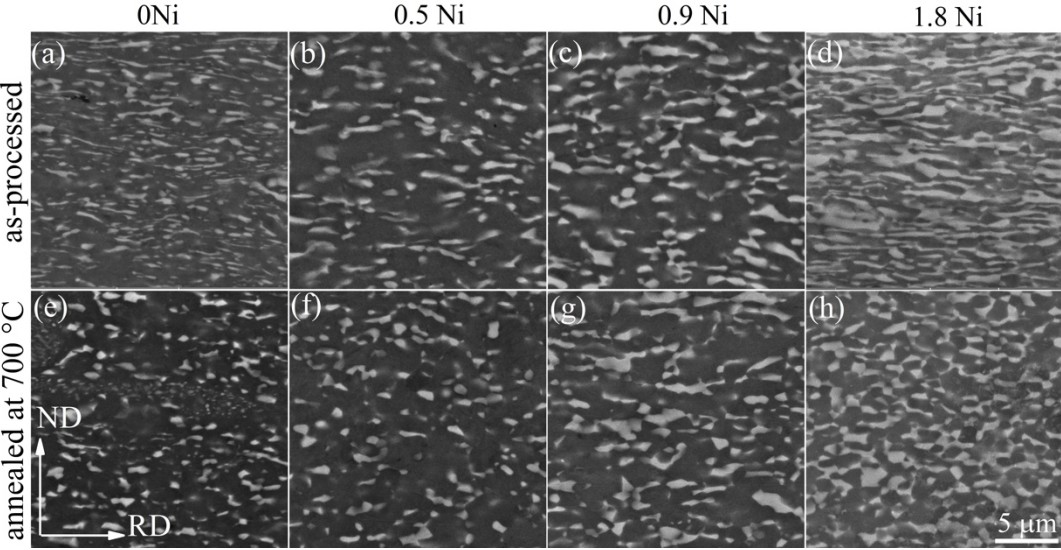

**Figure 2.** SEM images of (**a**–**d**) hot-rolled sheets at 750 °C and (**e**–**h**) annealed sheets at 700 °C (30 min) for the (**a,e**) 0Ni alloy, (**b,f**) 0.5Ni alloy, (**c,g**) 0.9Ni alloy, and (**d,h**) 1.8Ni alloy.

The α/β phase ratio depends on the rolling temperature, which influences the as-processed microstructure [2,42]. The hot-rolling temperature of 750 °C provided 20% of the β phase for the 0.5Ni alloy, 25% for the 0.9Ni alloy, and 35% for the 1.8Ni alloy. In addition, rolling was performed at a lower temperature of 700 °C to provide a similar β-phase fraction of ~20–25% for the 1.8Ni alloy (Figure 3). The 1.8Ni alloy demonstrated a significant difference in the microstructure after processing at 750 and 700 °C. The microstructure contained the α and β phases and bright particles with a mean size of 1.1 ± 0.1 μm and a volume fraction of ~1% (Figure 3a). The EDS analysis showed a high Ni content in these particles (Figure 3b). The XRD analysis in the as-rolled state detected the presence of the $Ti_2Ni$ phase in addition to the α, β, and TiB phases (Figure 3c). It should

be noted that the brittle intermetallic phases adversely influence the technological and mechanical properties of titanium alloys [30,43].

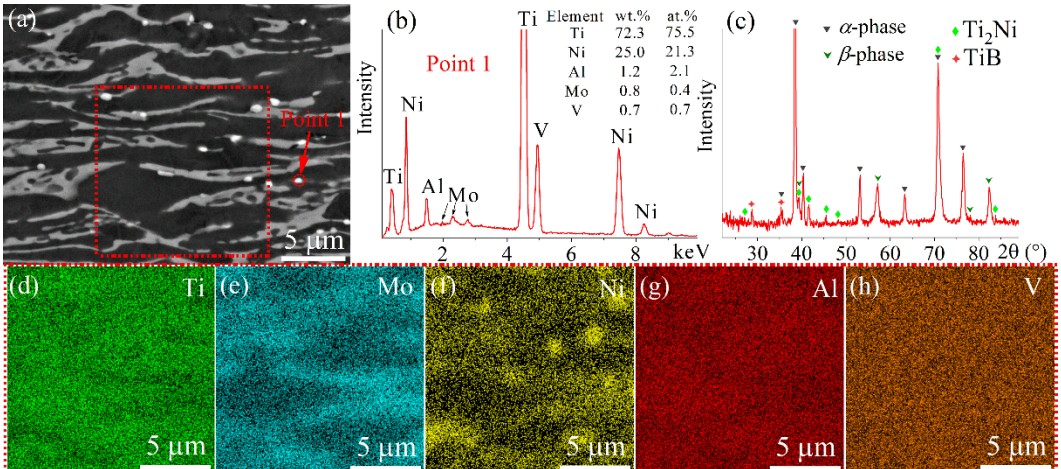

**Figure 3.** (**a**) SEM image, (**b**) EDS spectra for point 1 in (**a**,**c**) the XRD pattern, and (**d**–**h**) EDS-SEM maps of sheets of 2Ni alloy hot-rolled at 700 °C for the area marked with a dotted red frame in (**a**).

Grain sizes and volume fractions of the $\alpha$ and $\beta$ phases were measured after annealing at temperatures from 625 to 850 °C for the samples processed at 750 °C (Figure 4). As demonstrated by the $\beta$-phase volume fraction vs. temperature dependences, an increase in Ni content increased the volume fraction of the $\beta$ phase. Therefore, the temperature range of the optimal volume fraction of the $\beta$ phase (40–50%) shifted towards the lower temperatures with increasing Ni content (Figure 4a). The microstructure after annealing consisted of equiaxed fine grains (Figure 2e–h) with an average grain size in a range from 1.0 to 2.0 μm for the $\alpha$ phase and from 0.5 to 1.3 μm for the $\beta$ phase, which depended on the Ni content and annealing temperature. The modified alloys with 0.5–1.8 wt.%Ni exhibited a coarser $\alpha$-phase grain size (from 1 to 2 μm) compared to the Ni-free alloy (1.0–1.4 μm). On the contrary, an increase in the Ni content led to a slight decrease in the $\beta$-phase grain size (Figure 4b).

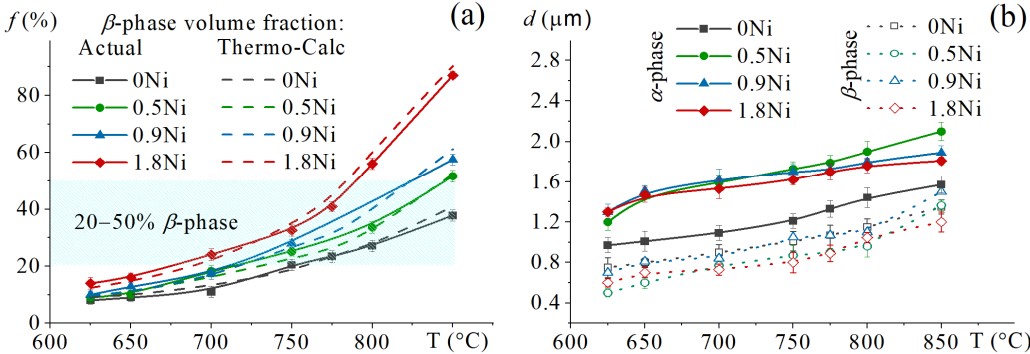

**Figure 4.** (**a**) Volume fraction (*f*) and (**b**) grain size of the $\alpha$ and $\beta$ phases vs. annealing temperature for the investigated alloys.

For the alloys with different Ni contents, the grain/subgrain structure was studied with EBSD analysis after annealing at a temperature of 700 °C for 30 min and water cooling (Figure 5). The $\beta$-phase fraction increased from 1 to 9% with an increase in the Ni content from 0.5 to 1.8 wt.% (green color in the EBSD maps in Figure 5a–c). These values were significantly lower than that of SEM BSE data and theoretical Thermo-Calc data. The BCC $\beta$ phase was partially transformed into the HCP $\alpha$ during cooling to room temperature. For the high-Ni alloy, a formation of a high fraction of the martensitic HCP $\alpha'$-phase was

possible owing to a decrease in the martensite start temperature and rapid cooling from the annealing treatment. As a result, the volume fraction of the β phase as determined with EBSD was lower than its high-temperature actual fraction as determined with SEM-BSE analysis. An increase in Ni content from 0.5 to 1.8 wt.% led to an increase in the α-phase mean grain size from $1.2 \pm 0.1$ to $1.9 \pm 0.1$ μm (considering both α and transformed β) and a slight decrease in the mean β-phase grain size from $0.9 \pm 0.1$ to $0.6 \pm 0.1$ μm, which agreed with SEM data analysis. A high fraction of low-angle grain boundaries (LAGBs) in the alloy with 1.8wt.% Ni (53% for 1.8Ni alloy compared to 26–27% for 0.5Ni and 0.9Ni alloys) may be the result of the martensitic phase formation owing to a decrease in the critical martensite start temperature [44].

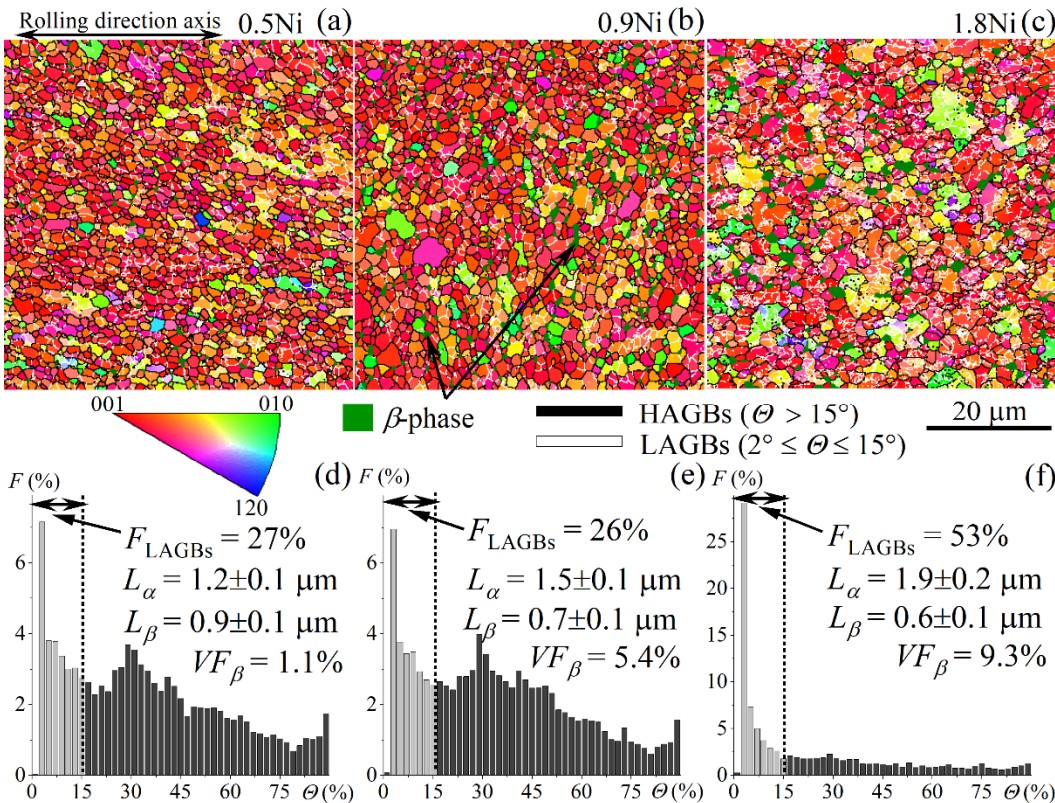

**Figure 5.** (**a–c**) Electron backscattered diffraction inverse pole figure (EBSD-IPF) grain boundary maps of the α phase and (**d–f**) the related grain boundary misorientation angle distributions for (**a,d**) the 0.5Ni, (**b,e**) 0.9Ni, and (**c,f**) 1.8Ni alloys after annealing for 30 min at 700 °C.

*3.3. Superplastic Deformation Behavior*

Figure 6a–c illustrate the sigmoidal-shaped stress–strain rate and strain rate sensitivity coefficient (*m*)–strain rate curves at the temperatures of 625, 700, and 775 °C. Alloying with Ni induced a decrease in the flow stress and an increase in *m*-value at all studied temperatures. The flow stress decreased, and the strain rate sensitivity coefficient (*m*) increased with an increase in the strain rate and temperature. The maximum *m*-values of 0.50–0.63 were obtained at a temperature of 775 °C with a strain rate range of $5 \times 10^{-5}$–$5 \times 10^{-3}$ s$^{-1}$ for the 0.5Ni, 0.9Ni, and 1.8Ni alloys, respectively, and the *m*-values were 0.40–0.43 for a strain rate range of $1 \times 10^{-4}$–$5 \times 10^{-3}$ s$^{-1}$ for the alloy 0Ni (Figure 6c). The 0.9Ni and 1.8Ni alloys exhibited a high m-value even for the low temperature of 700 °C (Figure 6b). The 1.8Ni alloy exhibited a high $m = 0.50$–$0.52$ at a temperature of 625 °C with a strain rate range of $5 \times 10^{-5}$–$1 \times 10^{-3}$ s$^{-1}$ (Figure 6a).

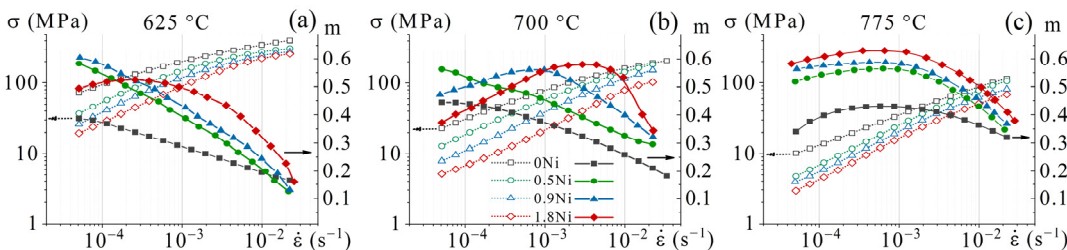

**Figure 6.** Stress vs. strain rate and *m* value vs. strain rate obtained by a step-by-step decrease in the strain rate at (**a**) 625 °C, (**b**) 700 °C, and (**c**) 775 °C.

Uniaxial constant strain rate tests were carried out at temperatures of 625, 700, and 775 °C and at a strain rate of $1 \times 10^{-3}$ s$^{-1}$ (Figure 7), which corresponded to a high *m* coefficient. Alloying with 0.9–1.8 wt.% Ni decreased flow stresses by 2–3-fold and increased the elongation to failure from 200–350% to 500–950%. The alloy with 0.5 wt.% Ni exhibited the maximum elongation of 700% at 775 °C. The deformation temperature of 775 °C provided the lowest stresses and the highest elongation to failure of the studied alloys. Figure 7c demonstrates the stress–strain curve for the 1.8Ni alloy hot-rolled at 700 °C. The elongation to failure decreased, and the flow stress increased as a result of the Ti$_2$Ni phase precipitation that occurred when the hot working temperature was lowered to 700 °C.

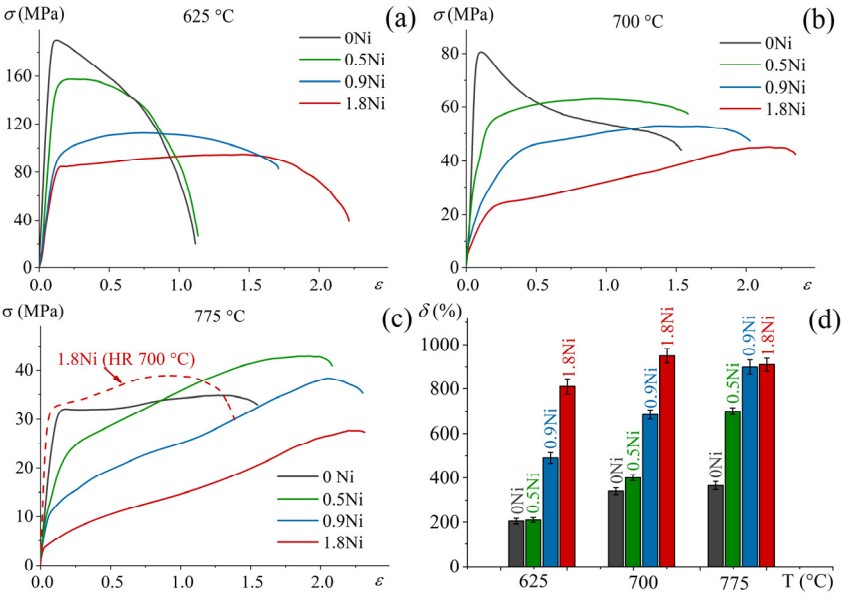

**Figure 7.** (**a**–**c**) Stress–strain curves at (**a**) 625 °C, (**b**) 700 °C, and (**c**) 775 °C and (**d**) dependency of elongation to failure ($\delta$) at a constant strain rate of $1 \times 10^{-3}$ s$^{-1}$.

The microstructural evolution during superplastic deformation has a strong influence on the superplastic behavior. Therefore, the strain-induced evolution of the strain rate sensitivity coefficient (*m*) at a nominal constant strain rate of $1 \times 10^{-3}$ s$^{-1}$ was investigated. The coefficient *m* increased from 0.3 to 0.4 with straining to 1.0 and decreased from 0.4 to 0.35 during further straining to 1.2 for the 0Ni alloy (Figure 8). The 0.5Ni and 0.9Ni alloys exhibited high *m*-values at the initial stage of deformation up to a strain of 1.0; 0.45–0.50 for the alloy with 0.5 wt.% Ni and 0.52–0.62 for the alloy with 0.9 wt.% Ni. A further strain increase led to a decrease in the *m* coefficient to 0.30–0.40 for the 0.9Ni alloy and to 0.39–0.48 for the 1.8Ni alloy. The 0.9Ni alloy demonstrated *m* values of about 0.35 throughout the tensile test at a low temperature of 625 °C (Figure 8a).

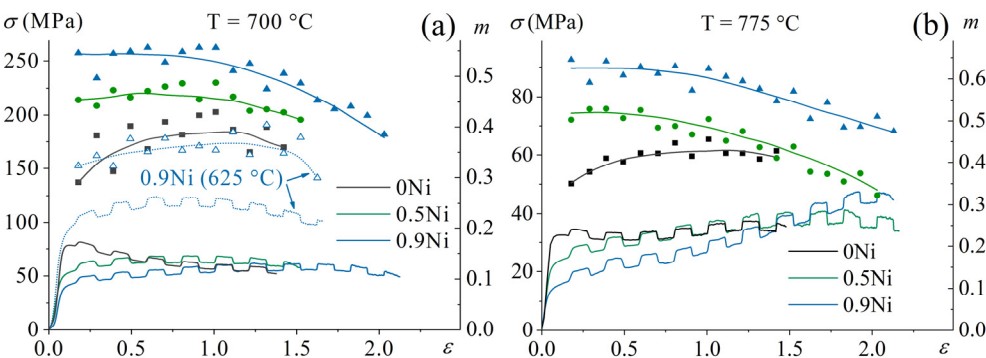

**Figure 8.** Stress–strain curves and coefficient *m* dependence of true strain during the step tests according to ASTM standard at (**a**) 700 °C and (**b**) 775 °C at the nominal strain rate of $1 \times 10^{-3}$ s$^{-1}$.

The following equation was used to calculate the activation energy of superplastic deformation (*Q*) after a strain of 0.69 (100% engineering strain) (Equation (1)) [45,46]:

$$Q = \frac{1}{m} \times R \times \frac{\partial \ln \sigma}{\partial 1/T}\bigg|_{\dot{\varepsilon}=const}, \tag{1}$$

where *R* is the universal gas constant (*R* = 8.314 KJ/mol), *m* is the strain rate sensitivity coefficient at the strain rate $1 \times 10^{-3}$ s$^{-1}$, $\sigma$ is the true stress for the 0.69 strain, and *T* is the temperature of superplastic deformation. The activation energies were 185 ± 8, 119 ± 8, 115 ± 9, and 132 ± 9 KJ/mol for 0Fe, 0.5Ni, 0.9Ni, and 1.8Ni alloys, respectively. Thus, the addition of 0.5–1.8 wt.% Ni led to a dramatic decrease in the activation energy. The activation energies of grain boundary self-diffusion for $\alpha$-Ti and $\beta$-Ti are 183 and 153 kJ/mol, respectively [26,45,47]. Thus, the *Q* values for the Ni-free alloy suggest the deformation is controlled by the self-diffusion at $\alpha$-Ti phase grain boundaries. For the Ni-bearing alloys, the superplastic deformation is controlled by the self-diffusion of $\beta$-Ti phase grain boundaries or Ni hetero-diffusion, for which the activation energy in $\beta$-Ti is 123.9 KJ/mol [48].

### 3.4. Microstructural Evolution during Superplastic Deformation

SEM micrographs at different true strains demonstrate an acceleration of the grain growth owing to Ni alloying (Figure 9). The grain sizes for the $\alpha$ and $\beta$ phases slightly increased during deformation at temperatures of 625–700 °C (Figure 9a–h). The difference in the grain sizes for the 0.5Ni, 0.9Ni, and 1.8Ni alloys was significant after the straining to 2.0 at 775 °C (Figure 9k–l). With an increase in the Ni content from 0.5 to 0.9 wt.%, and then to 1.8 wt.%, the $\alpha$-phase grain size grew from 2.6 ± 0.2 μm to 2.9 ± 0.2 μm and 3.4 ± 0.2 μm, and the $\beta$-phase grain size grew from 1.3 ± 0.1 μm to 1.6 ± 0.1 μm and 2.7 ± 0.1 μm, respectively.

The grain sizes were measured after the deformation at a constant strain rate of $1 \times 10^{-3}$ s$^{-1}$ and after annealing for a time corresponding to the tensile test time. The static and dynamic components of the grain growth during high-temperature deformation were determined. The static grain growth can be described via Equation (2) [49–52]:

$$d_s^n - d_0^n = K_s \times (t - t_0), \tag{2}$$

where $K_s$ is the constant of the static coarsening rate; $t_0$ is the annealing time, including the heating and holding time before the start of the tensile test, and in our case, it was 0.5 h; *t* is the staining time; $d_0$ is the initial grain size before the start of the deformation; $d_s$ is the grain size after annealing for a time *t*, and *n* is a static coarsening exponent.

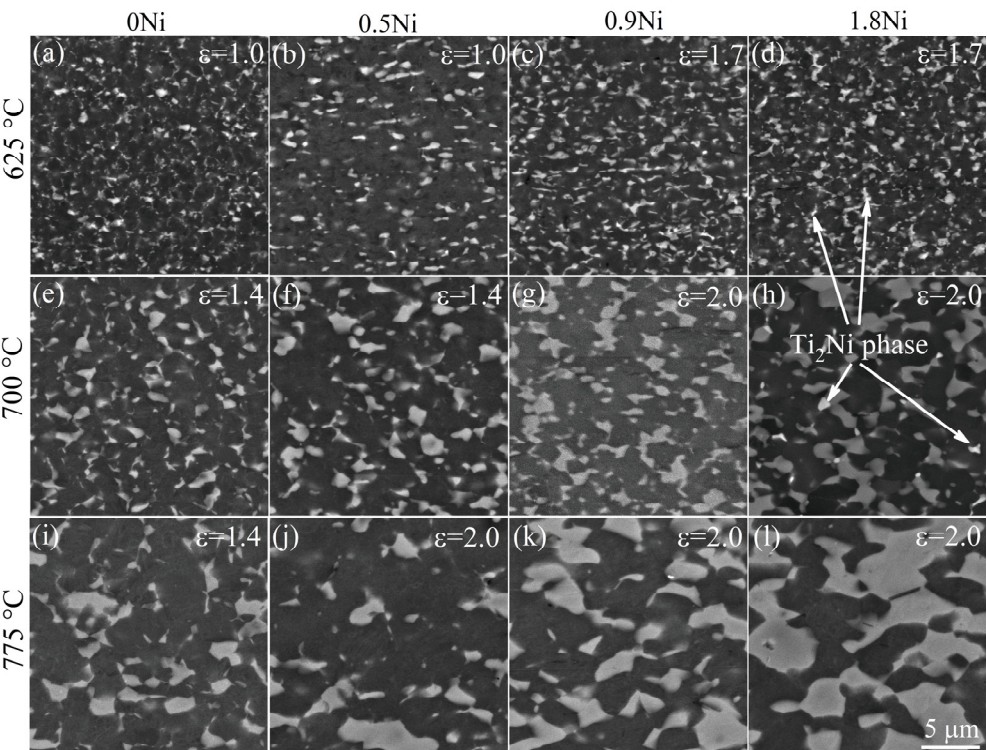

**Figure 9.** Microstructures of (**a,e,i**) 0Ni, (**b,f,j**) 0.5Ni, (**c,g,k**) 0.9Ni, and (**d,h,l**) 1.8Ni alloys after deformation at $1 \times 10^{-3}$ s$^{-1}$ at (**a–c**) 625 °C, (**e–h**) 700 °C, and (**i–l**) 775 °C with different true strains ($\varepsilon$, indicated on microstructures).

A similar equation was used to describe dynamic grain growth (Equation (3)) [49,52,53]:

$$d^n - d_s^n = K_d \times (t - t_0), \tag{3}$$

where $K_d$ is the constant of the dynamic coarsening rate; $d$ is the mean grain size after deformation with a strain corresponding to a tensile time $t$, and $n$ is the static coarsening exponent. Figure 10 contains the measured dependences of the grain size on the strain and deformation temperature for the studied alloys. The Ni-free alloy exhibited slight grain growth over the entire temperature range. On the contrary, an increase in the Ni content promoted grain growth, especially at a temperature of 775 °C.

The value of $n = 3$ for both static and dynamic grain growth was the closest to a linear relationship between $d^n - d_s^n$ and $t - t_0$ (Figure 11a,d). Based on these dependencies, the static and dynamic grain growth coefficients for $\alpha$ (Figure 11b,c) and $\beta$ (Figure 11e,f) phases were calculated. Static grain growth was insignificant for the alloys studied (Figure 11c,f). An increase in the Ni content strongly promoted the dynamic growth of the $\alpha$-phase grains at a temperature of 775 °C (Figure 11b). The dynamic $\beta$-phase grain growth was weaker, and it was observed only for the alloy with 1.8 wt.% Ni and at a temperature of 775 °C (Figure 11e).

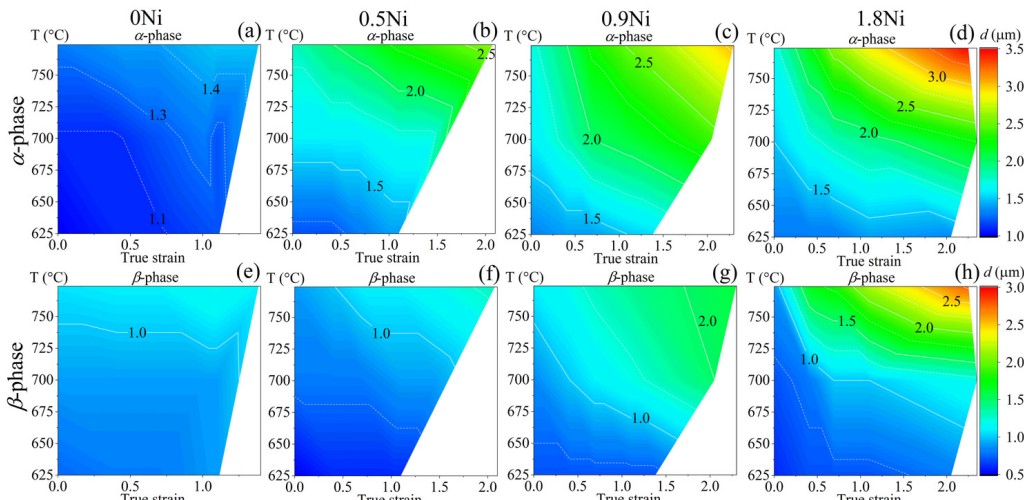

**Figure 10.** Grain size evolution of the (**a–d**) $\alpha$ and (**e–h**) $\beta$ phases vs. true strain of (**a,e**) 0Ni, (**b,f**) 0.5Ni, (**c,g**) 0.9Ni, and (**d,h**) 1.8Ni alloys during deformation at $1 \times 10^{-3}$ s$^{-1}$ and temperature range of 625–775 °C.

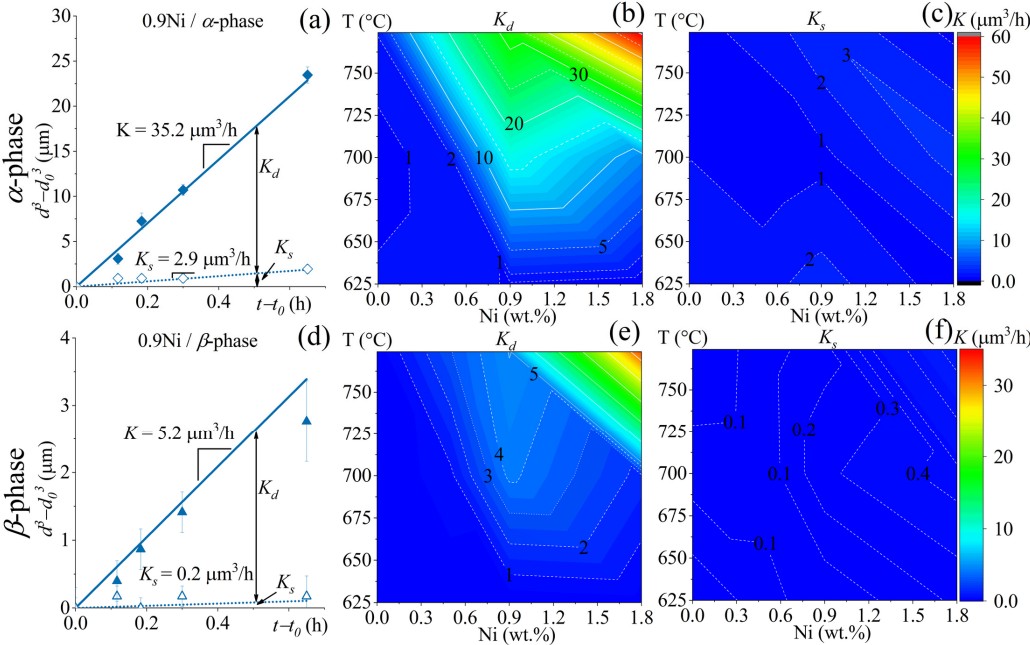

**Figure 11.** (**a,d**) Relationship of $d^3 - d_0^3$ vs. $t - t_0$ for 0.9Ni alloy and grain growth (**b,e**) dynamic ($K_d$) and (**c,f**) static ($K_s$) coefficients for (**a–c**) the $\alpha$ phase and (**d–f**) $\beta$ phase for the investigated alloys.

In the 1.8Ni alloy, the Ti$_2$Ni phase particles were observed after superplastic deformation at 625 and 700 °C (Figure 9d,h) similarly to the results obtained after hot rolling at 700 °C (Figure 3). It is notable that the Ni-enriched intermetallic phase was not found after annealing at the same temperatures (Figure 2h). The Ti$_2$Ni phase was also not observed after deformation at a temperature range of 750–775 °C. Therefore, the precipitation of the Ti$_2$Ni phase exhibited a strain-induced nature, and the phase precipitated at the low temperature deformation of 625–700 °C. The solvus temperature of the Ti$_2$Ni phase is 790 °C according to the Thermo-Calc data (Figure 1b). The predicted volume fractions of the Ti$_2$Ni phase were 2.9, 2.0, and 1.1% at the temperatures of 700, 750, and 775 °C, respectively. The measured volume fraction at 700 °C was only 1%. The difference between theoretical and experimental values could be the result of the non-equilibrium state of the alloys and requires further investigation.

Due to a high diffusivity in titanium and a $\beta$ stabilization effect, Ni increases a $\beta$-phase fraction and enhances the diffusion-controlled deformation mechanisms at a low deformation temperature that facilitates the GBS process and its accommodation; therefore, Ni improves low-temperature superplasticity. For the same reason of high diffusivity, Ni accelerated dynamic grain growth during superplastic deformation. An increase in the dynamic grain growth effect had a less significant influence on low-temperature superplasticity than an increase in alloy diffusivity had on the superplastic deformation mechanisms; therefore, superplastic elongations were improved. Meanwhile, 1.8 wt.% Ni led to a precipitation of the brittle intermetallic $Ti_2Ni$ compound during the deformation that decreased superplastic elongations. For the alloy with 0.9 wt.% Ni, the $Ti_2Ni$ phase was not precipitated, and good superplastic properties were observed at a temperature range of 625–700 °C. Therefore, the Ti-4Al-3Mo-1V-0.9Ni–0.1B (wt.%) alloy is the most promising for low-temperature superplastic forming.

### 3.5. Mechanical Properties

The mechanical properties at room temperature were determined for the 0Ni, 0.5Ni, and 0.9Ni alloys (Table 2). The mechanical properties were not determined for the 1.8Ni alloy, since undesirable $Ti_2Ni$ and $\omega$ phase formation is possible during the alloy processing and superplastic deformation. After a true strain of 0.69 (100% engineering strain) at a temperature of 775 °C, the samples were water-cooled and subsequently aged at 480 °C for 16 h. An increase in Ni content from 0 to 0.9 wt.% increased the yield strength from 740 to 885 MPa, increased the ultimate tensile strength from 865 to 1020 MPa, and slightly decreased the ductility from 8.1 to 6.6%. The increase in strength is associated with the solid solution strengthening [54,55] and the improved aging effect due to an increase in the $\alpha'$-phase fraction, which transforms into fine $\alpha$ and $\beta$ particles during aging treatment [56]. As a result, the alloy with 0.9 wt.% Ni is the most promising alloy due to it's low-temperature superplasticity and increased room temperature mechanical properties with a weak risk of strain-induced $Ti_2Ni$ phase precipitation.

**Table 2.** Yield strength ($\sigma_{0.2}$), ultimate tensile strength ($\sigma_u$), and elongation to failure ($\delta$) of the investigated alloys after superplastic deformation at 775 °C and 0.69 true strain (100% engineering strain) after quenching and aging.

| Alloy | $\sigma_{0.2}$ (MPa) | $\sigma_u$ (MPa) | $\delta$ (%) |
|---|---|---|---|
| 0Ni | 740 ± 5 | 865 ± 7 | 8.1 ± 0.5 |
| 0.5Ni | 790 ± 7 | 875 ± 8 | 6.4 ± 0.6 |
| 0.9Ni | 885 ± 8 | 1020 ± 6 | 6.6 ± 0.5 |

## 4. Conclusions

The influence of 0.5–1.8 wt.% Ni alloying on the microstructure evolution, superplastic deformation behavior in a temperature range of 625–775 °C, and mechanical properties at room temperature of the Ti-4Al-3Mo-1V-0.1B alloy was studied. The conclusions are summarized as follows:

1. Due to an increase in the $\beta$-phase fraction and alloy diffusivity, alloying with 0.5–1.8 wt.% Ni of the Ti-4Al-3Mo-1V-0.1B alloy substantially improved the superplastic characteristics; the strain rate sensitivity coefficient $m$ increased from 0.35–0.40 to 0.50–0.63, the flow stress decreased by 1.5–3-fold, and the elongation to failure increased from 200–350% to 500–950%.

2. Nickel insignificantly influenced the static grain growth but accelerated the dynamic grain growth during the superplastic deformation at a temperature range of 700–775 °C. An addition of 1.8 wt.% Ni promoted strain-induced $Ti_2Ni$ phase precipitation during hot rolling or superplastic deformation at a temperature range of 625–700 °C.

3. The 0.9 wt.% Ni addition increased the room-temperature mechanical strength; after superplastic deformation followed by quenching and aging, the ultimate tensile strength increased from 865 to 1020 MPa and the yield strength increased from 740 to 885 MPa. The elongation at fracture decreased from 8.1 to 6.6%. The Ti-4Al-3Mo-1V-0.9Ni-0.1B (wt.%) alloy has a promising composition that provides a good combination of low-temperature superplasticity and post-forming mechanical properties at room temperature.

**Author Contributions:** Methodology, data curation and investigation, M.N.P.; visualization and formal analysis, A.O.M. and V.V.C.; conceptualization, supervision, writing—review and editing, A.D.K. and A.V.M. All authors have read and agreed to the published version of the manuscript.

**Funding:** The study was funded by the Russian Science Foundation (Grant #21-79-10380).

**Institutional Review Board Statement:** Not applicable.

**Informed Consent Statement:** Not applicable.

**Data Availability Statement:** Not applicable.

**Acknowledgments:** 

**Conflicts of Interest:** The authors declare no conflict of interest.

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
