# Peer review of "Microstructure and Superplastic Behavior of Ni-Modified Ti-Al-Mo-V Alloys"

_metals, doi:10.3390/met12050741_

Round 1
Reviewer 1 Report
The manuscript reports the influence of the 0.5-1.8wt.% Ni alloying on the superplasticity, microstructural evolution in a low-temperature range of 625-775 °C, and room temperature mechanical properties of two-phase Ti-Al-Mo-V alloys. The article should be re-examined after revision. The issues are detailed below.
- The English should be carefully checked and improved. There are some problems such as wrong words and misuse of symbols in the article。
- The introduction is not clear, such as why Ni element is chosen as alloying elementfor this research?
- The title of Table 1 is incorrect.
- What is the detection limit of the rare phase by XRD?Can 1% Ti2Ni be detected? Verification is recommended.
- How is the statistic of deformed grainin Fig.5?
- The text in line 225-227 is inconsistent with the picture data, the author is suggested to verify.
- The author mentioned that “The value of n = 3 for both static and dynamic grain growth was the closest to a linear relationshi”, Please state that how the value of n is chosed or caculated ?
- It is stated in the paper that Ti2Ni precipitates are not observed in the annealing microstructure after rolling, but the conclusion is put forward that the stress duringthe rolling or tensile deformation induces the precipitation of Ti2Ni phase. There are certain contradictions, and it is suggested to verify. In addition, it is suggested to discuss why there is no precipitate phase during the tensile deformation at 775℃.
Author Response
We thank the Reviewer for their insightful comments on our manuscript. We have taken all the comments into consideration, and we have marked changes in the manuscript with blue-colored text and the changes are presented step-by-step in this detailed response. We carefully checked the English writing to improve the language of the manuscript.
Please see the attachment.

Reviewer 2 Report
This manuscript presents an interesting study of the superplasticity of a Ti-Al-V-Mo alloy. A high tensile elongation of 500-900% at 625-750 degree C was obtained by alloying 0.9% Ni to the alloy. Meanwhile, the alloy also showed good mechanical properties at room temperature. Therefore, the present study can direct the development of new grade Ti alloys. However, the results were not well discussed. Besides, there are several points need to be clarified:
- Boron element was added to the alloy. What is the purpose? The precipitated phase is TiB or TiB2? Please check the figure (Fig. 1) and related context. Since there is boron bearing precipitation in all samples, what is the effect on the superplasticity?
- Nickle promotes the dynamic grain growth and the generation of low angle grain boundaries of the alloy (Fig. 5 a-c). Is such effect favorable or unfavorable to the superplasticity? What is the mechanism for the superplasticity of the present alloy?
- Language need careful proof reading:
Line 47, 40-50%?
Line 128, ThermoCalc?
Line 187, (beta)-phase?
Author Response

(The authors gave the same response as above.)

Reviewer 3 Report
The submitted manuscript discusses the micorstructure evolution and superplastic properties of Ni-modified Ti-Al-Mo-V alloy. The manuscript falls within the scope of the journal. The English writing needs more polishing to improve the manuscript language.
After careful l reading, the reviewer made some general comments and specific comments can be found in the attached file. Authors are kindly requested to answer all queries in the attached file in order to improve the quality of the submitted manuscript.
1- The caption of Table 1 is incorrect. Authors are requsted to revise the subbmitted manuscript.
2- The initial diemnsions of the processed specimen (prior rolling) is not mentioned in the manuscript. Authors must provide this missing information.
3- In the XRD analysis of Figure 1, it is strange to not detect TiB2 peaks, although the phase was found in the calculations in the same figure.
4- The Ni content in Table 1 was 0, 0.5, 0.9 and 1.8% the reviewer is wondering why in the entire paper the Ni content was mentioned as 0, 0.5, 1 and 2? Authors must clarify or correct this misleading information.
5- There is a problem in using the scientific terms. For instance, the authors used the term grain size to describe the size of the phase precipitate. Another mistake is using the term HSP instead of HCP to describe the structure of a certain phase.
6- The SEM images did not show the grains. They only showed the phases. Authors can revise the attached file to find the location of this comment in the submitted manuscript.
7- The data in Section 3.2 is manuplated and incorrectly mentioned. The phase size did not increase proportionally with Ni content. This comment can be found more clearly in the attached file.
8- The results of EDS elemental mapping must be revised and the presence of Ti2Ni is not well ocnfirmed.
9- The remaining comments can be found in the attached file.

Author Response

(The authors gave the same response as above.)

Round 2
Reviewer 2 Report
The revision is fine. Can be accepted.
Reviewer 3 Report
The authors answered all comments sufficiently and the revised manuscript can be accepted in the current form.